# Novel Knowledge about Molecular Mechanisms of Heparin-Induced Thrombocytopenia Type II and Treatment Targets

**DOI:** 10.3390/ijms24098217

**Published:** 2023-05-04

**Authors:** Aušra Mongirdienė, Agnė Liuizė, Artūras Kašauskas

**Affiliations:** 1Department of Biochemistry, Medical Academy, Lithuanian University of Health Sciences, LT-50161 Kaunas, Lithuania; 2Medicine Academy, Lithuanian University of Health Sciences, Eiveniu Str. 4, LT-50103 Kaunas, Lithuania; agne.abramaviciute@stud.lsmu.lt

**Keywords:** heparin-induced thrombocytopenia (HIT), PF4-heparin antibody complexes, FcγRIIa receptors, monocytes, endotheliocytes, neutrophils, NETosis, PSGL-1, Syk kinase

## Abstract

Heparin-induced thrombocytopenia type II (HIT II), as stated in the literature, occurs in about 3% of all patients and in 0.1–5% of surgical patients. Thrombosis develops in 20–64% of patients with HIT. The mortality rate in HIT II has not decreased using non-heparin treatment with anticoagulants such as argatroban and lepirudin. An improved understanding of the pathophysiology of HIT may help identify targeted therapies to prevent thrombosis without subjecting patients to the risk of intense anticoagulation. The review will summarize the current knowledge about the pathogenesis of HIT II, potential new therapeutic targets related to it, and new treatments being developed. HIT II pathogenesis involves multi-step immune-mediated pathways dependent on the ratio of PF4/heparin and platelet, monocyte, neutrophil, and endothelium activation. For years, only platelets were known to take part in HIT II development. A few years ago, specific receptors and signal-induced pathways in monocytes, neutrophils and endothelium were revealed. It had been shown that the cells that had become active realised different newly formed compounds (platelet-released TF, TNFα, NAP2, CXCL-7, ENA-78, platelet-derived microparticles; monocytes-TF-MPs; neutrophils-NETs), leading to additional cell activation and consequently thrombin generation, resulting in thrombosis. Knowledge about FcγIIa receptors on platelets, monocytes, neutrophils and FcγIIIa on endothelium, chemokine (CXCR-2), and PSGL-1 receptors on neutrophils could allow for the development of a new non-anticoagulant treatment for HIT II. IgG degradation, Syk kinase and NETosis inhibition are in the field of developing new treatment possibilities too. Accordingly, IdeS and DNases-related pathways should be investigated for better understanding of HIT pathogenesis and the possibilities of being the HIT II treatment targets.

## 1. Introduction

Heparin-induced thrombocytopenia (HIT) is a disorder where, despite decreasing platelet count, thrombosis can develop [1]. According to its clinical features, HIT is divided into two types: HIT I and HIT II [2]. HIT I is likely to be non-immune and depends on a heparin platelet proaggregating effect [3]. HIT I is also known as heparin-associated thrombocytopenia (HAT). HIT I frequency is up to 10 % and is much more common than type II: [2,4]. HIT I patients usually do not have thrombosis, thrombocytopenia is mild, occurs usually within 2 days after heparin therapy [5] and resolves even without stopping the heparin therapy [6]. HIT I will not be analysed in this paper. The development of HIT II depends on an immune mechanism [7]. HIT II, as stated in the literature, occurs in about 3% of all patients and in 0.1–5% of surgical patients [2,8]. In patients receiving low molecular weight heparin (LMWH), the HIT ratio is 0.1–0.5% [9,10]. Thrombosis develops in 20–64% of patients with HIT [11,12]. The mortality rate in HIT II has not decreased using non-heparin treatment with anticoagulants such as argatroban and lepirudin [13]. The presence of HIT was associated with a 50% increase in perioperative mortality [8]. Postoperative outcomes such as acute renal failure, major amputation, need for tracheostomy, acute respiratory failure and stroke are more common in patients who developed HIT II after surgery and prolonged hospital stay [8]. Facts of spontaneous HIT (when patients develop symptoms of HIT without treatment with heparin) have been reported too [14,15]. Its pathogenesis is well reviewed by A. Greinacher and others [16] and will not be discussed here.

An improved understanding of the pathophysiology of HIT may help identify targeted therapies to prevent thrombosis without subjecting patients to the risk of intense anticoagulation. The review summarizes the newest knowledge about the pathogenesis of HIT II and the new treatment being developed related to it. 

## 2. Pathogenesis of HIT II

It is known that HIT II develops because platelet factor 4 (PF4)-heparin complex formation leads to antibody synthesis [7]. PF4 is a chemokine released from platelet α-granules [17]. Low levels of PF4 can be found in the blood but it increases after platelet activation [18]. Cationic PF4 binds to anionic heparin due to the electrostatic interactions [18,19]. The number of PF4-heparin complexes formed was found to be maximal at the equimolar PF4/heparin ratio [17,20]. 

PF4-heparin complexes were shown to activate complement, resulting in C3/C4 deposition on the complexes. Complement deposition on the complexes allows them to bind to B cells by the complement receptor 2 (CR2/CD21), leading to its activation and antibody synthesis [21]. The complement deposition on platelets could participate in platelet activation too (Figure 1). Complement, as it is presented in the literature, can be activated by three pathways. The classical pathway is one of the pathways [22], and IgM have been shown to take place in complement activation by the classical pathway. IgM binds to PF4-heparin complexes and undergoes a conformational change leading to activation of C1, activation of the C3 convertase, incorporation of the C3 fragments into PF4-heparin complexes, subsequent binding of IgM-C3 and antigen deposition on B cells via CR2/CD21 [23]. It was shown that disruption of IgM-C1 interaction could prevent PF4/heparin-mediated complement activation [23]. It could be a therapeutic target in HIT II. It is not clear what concentration of the IgM and/or PF4-heparin initiates complement activation and why complement activation by PF4-heparin complexes varies widely among healthy persons.

In addition, it was found that heparin (polysaccharide) binding to PF4 (protein) induces protein conformation changes that form sites for antibody binding [24]. PF4-heparin complex could bind to a platelet via the heparin-binding sites [19] by their heparin component or by their PF4 component. PF4 binding to heparin was shown to depend on polysaccharide chain length (the minimum length for polysaccharide-heparin binding was found to be six disaccharide subunits) [17,23,25] and sulphation degree [26]. Antibodies binding to PF4-heparin complexes on platelets result in platelet activation and aggregation [20,26], leading to the activation of coagulation pathways and thrombus formation (Figure 1). 

The reason PR4-heparin antibodies lead to thrombocytopenia and thrombosis in some patients but not in others is not known [27]. In HIT II, IgG antibodies to PF4-heparin complexes are thought to activate platelets, leading to thrombosis [28]. IgG antibodies, but not IgA or IgM [8], bind to platelet FcγRIIa receptors (CD32, the low-affinity IgG receptor) and provoke platelet activation [27]. The immune complex connection to FcγRIIa initiates phosphorylation of the immunoreceptor tyrosine-based activation motif (ITAM), leading to activation of downstream signalling via spleen tyrosine kinase (Syk) with the attendant release of intracellular Ca^2+^ stores, degranulation, cytokine production and cellular activation [29]. Platelet activation leads to (1) degranulation resulting in an increase in thromboxane A2 [30] and PF4 concentration and further antigen-complex formation [27], (2) release of serotonin [31], upregulation of P-selectin that takes part in the formation of platelet-leukocytes aggregates [32] and (3) production of platelet-derived microparticles, which act as a procoagulant [33]. All these processes lead to the activation of platelet and blood plasma coagulation factors. The interaction of IgG with the FcγRIIa receptor needs a specific region in IgG. It was found that an IgG-degrading enzyme of *Streptococcus pyogenes* (IdeS) would cleave off the whole Fc fragment at a specific site in the mentioned region and disable platelet activation via this pathway [34]. IdeS has been studied in immune thrombocytopenia and IgG-HIT mice models [34] and have shown promising results but still needs to be more widely investigated in humans. In vivo studies have shown the efficacy of the Syk kinase inhibitor (PRT-060318 (PRT318)) validating the importance of the FcγRIIa signalling pathway for prevention of thrombosis in HIT [35].

The immunogenicity of the PF4-heparin complexes depends on their net charge, and a positive net charge was shown to facilitate the interaction with immune cells [21]. It was later discovered that PF4 binds to glycosaminoglycans (GAGs) not only on platelets [36] but on monocytes [37], neutrophils [13,38] and endothelial cells too [13]. 

It was shown that thrombosis and thrombocytopenia in HIT are two separate processes, both depending on HIT immune complex activity [28]. Complexes bound to platelets via FcƴRIIa in the absence of neutrophil extracellular traps–NETosis resulted in thrombocytopenia [28]. NET are extruded strands of DNA, a granular material that serves critical functions in innate immunity by attracting pathogens [39]. NET was found to bind the PF4, directly affect platelets and erythrocytes [40], and activate the thrombin generation [41]. Therefore, thrombosis in HIT develops because of platelet activation, leading to activation of different cell types: the hypercoagulable state in HIT directly depends on the platelet [27], monocyte [42] and neutrophil activation through FcƴRIIa receptors on these cells and indirectly on activation of endothelial cells through different mechanisms [13]. 

To summarise, PF4-heparin complexes are the main trigger of HIT. Thrombosis complications develop along the same ways by which these complexes act: (1) activate complement and bind to it, leading to PF4-heparin-complement complexes binding to B cells and resulting in antibody IgG synthesis; (2) antibodies bind the PF4-heparin complexes and connect to platelets, monocytes, neutrophils and endotheliocytes through FcγRIIa receptors activating all these cells; (3) the activated cells secrete compounds that further increase activation of the cells and plasma coagulation factors, leading to cell aggregation and prothrombin activation and finally resulting in thrombosis. Three new potential treatment targets were found: (A) signal transduction through FcγRIIa receptor messenger Syk inhibition; (B) IgG degradation by IdeS, when degrading IgG cannot interact with FcγRIIa receptors and signal transmission is stopped; (C) IgM-C1 interaction, when complement can not be activated and affect the platelet. Inhibition of platelet CD21(complement receptors) could be investigated as a possible treatment target too.

### 2.1. Neutrophils in HIT Thrombosis Formation

It is thought that neutrophils take part in heparin-induced thrombosis. Neutrophils were shown to promote NET formation and thrombosis in HIT [28,43,44] through FcγRIIa receptors. Activation of neutrophils through FcγRIIa receptors [28,43,45] (and through P-selectin) leads to the generation of reactive oxygen species, degranulation [28,32,43,46] increased phagocytosis [44], integrin expression [38] and formation of neutrophil-platelet aggregates [29]. PF4-heparin antibodies bind to PF4 bound to chondroitin sulphate on neutrophil surfaces [41]. It was shown that FcγRIIa receptors on neutrophils binding to antibodies enhanced neutrophil aggregate formation through increased expression of macrophage adhesion molecule-1 (oligodeoxynucleotide-binding protein CD11b (Mac-1)) on the neutrophil surface [38] (Figure 2). Additionally, myeloperoxidase (MPO) levels were found to be higher in patients with HIT [29], suggesting HIT association with neutrophil activation. Kandace Gollom with the research group revealed steps in HIT by which neutrophils participate in thrombosis: (1) adhesion of stimulated neutrophils to inflamed endothelium, (2) the promotion of neutrophil migration into venular thrombi, (3) formation of antigenic PF4-(NET)-HIT antibody complexes [40,43,44]. 

A few events could enhance neutrophil adherence to the endothelium. The first could be HIT antibody exposure increasing the Mac-1 expression on the surface of neutrophils [38]. The second one is believed to be the growing venular injury because of the increased turbulent flow around the thrombi [38]. The third, activation of endothelial cells by HIT antibodies, increases the expression of P and E selectins [47]. During thrombus formation, PF4 release from platelets within the thrombus could lead to an increased assembly of antigenic complexes, resulting in more HIT antibodies binding to endothelial cells and their activation [13]. Additionally, platelets undergo degranulation when forming a thrombus in the veins. After entering the blood, platelets release neutrophil-activating peptide 2 (NAP2 or CXCL7) and epithelial-derived neutrophil-activating peptide 78 (ENA-78 or CXCL5), leading to chemokine receptor CXCR2-dependent neutrophil activation [48]. Activated neutrophils release NETs [28,43,49] and form complexes with PF4 that are subsequently bound by HIT antibodies [40]. Additionally, HIT Ig G activates neutrophils to induce NETosis [28,43]. The concentrations of neutrophil activation markers (MPO, elastase, extracellular DNA and citrullinated histones (CitH3)), NET neutrophils and neutrophil–platelet aggregates were shown to be higher in HIT patients [28,40,43]. Neutrophils with nuclear decondensation (NETosis marker), citrullinated histone H3 (CitH3) and extracellular DNR were found in platelet-rich thrombi in the lungs of mice with HIT [28,43]. PF4-NET-HIT antibody complexes were shown to be protected from degradation by DNases (a deoxyribonuclease—an enzyme that catalyses the hydrolytic cleavage of phosphodiester linkages in the DNA backbone, thus degrading DNA) and could contribute to thrombus development on a Fc-rich surface [40]. Formation of smaller venous thrombi has been shown in HIT mice models when neutrophils could not undergo NETosis but enhanced arterial clots, and thus thrombocytopenia developed [40]. DNase treatment could prevent the increase in venular thrombus size and prothrombotic state in HIT. Neutrophil migration into venous thrombi could be stopped by the blockade of CXCR2 [40]. However, the NAP2 and ENA-78 contribution to the chemo gradient should be investigated further.

It was found that immune complexes (ICs) could activate neutrophils in HIT. Activation goes via FcƴRIIa: (1) directly, and (2) by platelet activation leading to P-selectin and P-selectin glycoprotein ligand-1 (PSGL-1) release from activated platelets resulting in neutrophil activation [28,43]. Despite the P-selectin/PSGL-1 interaction not being the only mediator of the neutrophil/platelet interaction [40,50], the blockade of P-selectin (CD62p) or PSLG-1 (CD162) markedly inhibits neutrophil/platelet interaction induced by HIT antibodies and prevents DNA release by neutrophils in the presence of HIT ICs activated platelets [28]. Therefore, it seems that neutrophil/platelet interaction through CD62p and CD 162 is very important in HIT thrombosis and could be a treatment target for avoiding the thrombosis complications in HIT. 

Furthermore, it was shown in the HIT model using FcƴRIIa^+^/hPF4^+^ mice that treatment with DNase I lowered thrombus formation, and peptidyl Arginine Deiminase 4 (PAD4) inhibitor GSK 484 ((3S,4R)-3-Amino-4-hydroxypiperidin-1-yl)(2-(1-(cyclopropylmethyl)-1H-indol-2-yl)-7-methoxy-1-methyl-1H-benzo[d]imidazol-5-yl)methanone) stopped NETosis and decreased thrombus formation significantly [28]. It is worth mentioning that thrombi failed to develop in PAD4-deficient mice when they were treated with HIT antibodies and heparin [28]. Additionally, the size of thrombi was shown to depend on NETosis [28]. NETosis may partially explain treatment with anticoagulants alone being insufficient for thrombosis in HIT. 

Summarizing the importance of neutrophils in the course of HIT, it can be said that neutrophils promote thrombosis through NET formation and endothelium activation. Neutrophils can be activated (1) through FcƴRIIa receptors by PF4-heparin-IgG complexes, (2) through CXCR2 and PSGL-1 receptors by activated platelet secreted compounds, or (3) through GAGs on the neutrophil surface by PF4-heparin-IgG complexes (these complexes were found to be protected from degradation by DNases). Activated neutrophils release NETs that consequently activate prothrombin and lead to thrombosis. Therefore, NETosis inhibition could be one of the treatment targets for thrombosis reduction in HIT. It seems that NETosis can be inhibited by DNase treatment and PSLG-1 blockade.

### 2.2. Monocytes in HIT Thrombosis Formation

Monocytes take part in thrombus generation in HIT II as well because they interact with PF4-heparin antibody complexes through Fcy receptors, leading to tissue factor (TF) production [46]. PF4 binds to monocytes with higher affinity than they bind to platelets because of different GAGs in their glycocalyx. PF4 affinity for the different GAGs appears to be variable. Additionally, PF4 could bind to chondroitin-4-sulphate, chondroitin-6-sulphate, dermatan sulphate, heparan sulphate and heparin with increasing intensity [51]. In vivo studies in mice showed the moderating effect of differential binding of ultra large HIT immune complexes (ULICs) to platelets and monocytes on thrombocytopenia [39]. After heparin binds to PF4 on the monocyte, antibody/antigen complexes form and result in monocyte activation [52] leading to (1) tissue factor (TF) expression [1,42], (2) production of TF-expressing microparticles (TF-MPs), (3) IL-8 secretion [42], and (4) monocyte glycocalyx sulfation that induces a higher affinity for PF4 binding [53]. TF expression is shown to be dependent on FcƴRIIa too (1). Monocyte activation was additionally shown to depend on downstream signalling through Syk kinases (1). Inhibition of the Syk kinase pathway was shown to markedly reduce thrombin and fibrin formation (1). TF-MPs production and synthesis depend on FcƴRI [54]. All mentioned monocyte changes take place in procoagulation increase by thrombin generation [53]. Thrombin activates platelets via FcƴRIIa [54]. Monocytes are shown to be activated by P-selectin expressed by platelets. It should be mentioned that TF generated from monocytes activates platelet protease-activated receptors (PAR-1) [55]. Platelet activation through PAR-1 and FcƴRIIA [56] (or thrombin and collagen, or thrombin and FcƴRIIA) leads to the generation of a subpopulation of highly activated platelets named coated platelets [1]. Monocyte activation by HIT ULICs leads to Mac-1 expression and, consequently, platelet-monocyte aggregation [32]. It should be mentioned that IdeS treatment abolishes the increase in TF expression induced by HIT antibodies [33] and could also reduce thrombotic risk.

Summarizing, monocytes can be activated by PF4-heparin antibody complexes binding to FcyR IIa receptors (leading to Mac-1 expression and resulting in platelet–monocyte aggregation), platelet-expressed P-selectin, or GAGs on the platelet surface. Activated monocytes produce TF-MPs and other compounds related to the development of thrombosis. The pathways of monocyte activation through FcyR IIa receptors were shown to be realized through Syk kinases. Thus, Syk kinase inhibition affecting both monocytes and platelets could be used for thrombosis prevention.

### 2.3. Endothelium in HIT Thrombosis Formation

The endothelium could promote thrombosis in HIT by direct or indirect cellular injury. Surgery, catheterization, or underlying atherosclerosis promotes vWF release, which supports platelet binding sites for HIT antibodies. Complement activation and products of other activated cells through FcƴRIIA could activate endothelium indirectly leading to the expression of TF and adhesion markers. HIT antibodies bind the endotheliocytes causing immune injury [57]. It seems that HIT antibodies bind to PF4 on endothelium via the F(ab) region on the antibodies and need an endothelial preactivation, which is thought to depend on TNFα released from activated platelets [47]. Endothelial preactivation could be achieved after its injury, when vWF is released, leading to the increase in PF4 binding to vWF strings and resulting in the expression of antigenic sites for HIT antibodies [58]. HIT patient antibodies were shown to bind directly to endothelium cells via cellular GAGs [59]. In regions of vascular lesions, PF4 can bind to endothelium. Antigen–antibody complexes activate endothelium [47,58], leading to (1) changes in the glycocalyx of these cells [60], and (2) expression of adhesion molecules [47] and TF [60]. The PF4 could bind to strings of vWF released by injured or activated endothelium [61]. Antibodies from PF4-heparin complexes can bind to the endothelial cells after vessel injury [13]. The HIT antigen adheres to the bound PF4. The complex is recognised by HIT antibodies resulting in complement and the activation of other cells. 

In summary, endothelium could be activated by direct cellular injury or indirectly by the complement system, PF4-heparin-IgG complexes through FcγRII A, GAGs, and products of other activated cells. HIT antibodies binding to endothelium requires preactivation. Preactivation depends on TNFα released from activated platelets or direct injury. Activated endothelium releases TF and expresses adhesion molecules. FcγRII A blockers seem to be useful for protection against thrombosis by acting on endothelium.

## 3. HIT Prevention and Peculiarities of HIT Treatment

To avoid HIT-induced thrombosis, preoperative testing for plasma concentration of heparin-PF4 antibodies is offered to two groups of patients. The first group is patients with HIT evidenced by a drop in the platelet count or development of a new thrombotic event with a clear temporal association to heparin administration. The second group consists of patients who have a history of HIT and will be undergoing cardiac surgery with plans for intraoperative use of heparin [8]. It should be emphasized that high antibody levels or heightened platelet activity are the risk factors of thrombosis. To prevent thrombosis, it is also important to clarify the risk factors described in Section 3. 

Despite discrepancies among the various clinical practice guidelines for HIT [62,63,64], the lowering of thrombin generation is stated as the main aim of HIT treatment [62]. Direct thrombin inhibitors, heparin derivatives, and direct oral anticoagulants are used. Observation data suggest that direct oral anticoagulant usage for HIT treatment has not significantly affected the incidence of HIT [9]. Cardiac surgery patients exposed to heparin remain at considerable risk for HIT (about 3%) [8,10]. It is difficult to predict the outcome in patients undergoing surgery with a history of HIT. According to their condition, patients could be divided into two groups: (A) patients with acute HIT or persisting positive functional assay and PF4-heparin antibodies, and (B) patients with negative functional assay and persisting PF4-heparin antibodies. Surgery is considered to be safe if circulating PF4-heparin antibodies are no longer detectible [65]. The American Society of Haematology (ASH) recommends delaying cardiovascular surgery for patients of group A [62]. If the condition of these patients requires immediate surgery, the following is recommended: (1) intraoperative anti-aggregation (iloprost or tirofiban) and simultaneous anticoagulation with heparin, (2) alternative intraoperative anticoagulation with bivalirudin, or (3) peri-operative plasma exchanges and anticoagulation with heparin [62]. 

Safe usage of intraoperative heparin and simultaneous cangrelor was reported in patients of B group, but subsequent rising of PF4-heparin antibodies and positive functional assay seroconversion was found even a short heparin prescription (<1.5 h) [66]. 

Direct thrombin inhibitors (argatroban, bivalirudin), indirect factor Xa inhibitors (fondaparinux, danaparoid) and desirudin (direct thrombin inhibitor) are direct action nonheparin anticoagulants used in the treatment of HIT [67]. High-quality prospective head-to-head trials comparing these agents could not be found. Bivalirudin is used during coronary interventions and cardiac surgeries in patients with and without HIT, although it is not approved by the FDA exclusively for HIT [68]. The CHEST guidelines support the use of bivalirudin over argatroban in patients with HIT who need coronary interventions. Bivalirudin has a short half-life of 25 min and has enzymatic as well as renal clearance. Due to its moderate elevation of PT/INR compared to argatroban, bivalirudin offers an easier transition to an oral anticoagulant such as warfarin [69]. However, the data on direct oral anticoagulant usage in HIT with thrombosis and acute HIT remains limited [67]. 

Tirofiban (fibrinogen receptor blocker) has 1.4–2.2 h half-life and depends on renal function [70]. Iloprost (synthetic analogue of PGI2 inhibiting platelet granule release, adhesion, and aggregation) has 30 min half-life [71]. Cangrelor is a rapid-acting reversible ADP receptor P2Y12 inhibitor with a half-life of 3–6 min [72]. It was unable to inhibit heparin-induced platelet aggregation in vitro in the presence of PF4-heparin antibodies [73], despite the drug efficacy shown in a particular patient with a pre-surgery negative aggregation test [73].

Non-anticoagulant therapy involving removal of anti-PF4-heparin antibodies [74] and inhibition of antibody-mediated platelet activation [75], rapid-acting reversible platelet inhibition by cangrelor [66,73] and endopeptidase specifically cleaving IgG antibodies (imlifidase) [36] have been reported for HIT treatment recently. Onuoha with co-authors reviewed achievements of plasmapheresis for HIT treatment [74]. He concluded that despite TPE being commonly used in patients undergoing cardiopulmonary bypass surgery (CPB), prospective studies were needed to clarify which treatment (intravenous immune globulin (IVIG) or therapeutic plasma exchange (TPE)) is indicated in HIT population subsets. Intravenous immunoglobulin (IVIG) treatment was defined as a research priority for the HIT treatment [62]. It is believed that IVIG inhibits platelet and other cell activations through FcγIIa receptors. Several studies have shown that it competes with anti-PF4/H to bind to the FcgRIIA receptor depending on the FcgRIIA H31R polymorphism, but the exact mechanism of IVIG remains unknown [75]. Cangrelor has been shown to be useful in cardiovascular surgery when PF4-heparin antibodies are present and intraoperative heparin use is mandatory [66,73]. However, the treatment efficacy must be assessed for each patient with a functional assay before its use [73]. IdeS has not yet been studied in HIT patients. 

All medications recognized and used instead of heparin inhibit the clotting process and are therefore dangerous due to the risk of bleeding. Therefore, new classes of drugs are being sought to avoid these side effects. Removal of anti-PF4-heparin antibodies and inhibition of antibody-mediated platelet activation, rapid-acting reversible platelet inhibition by cangrelor and endopeptidase specifically cleaving IgG antibodies (imlifidase), and IdeS have been investigated for HIT treatment recently. New research into the pathogenesis of HIT aims to find other new treatment possibilities. Recent studies have identified some sites of HIT pathogenesis that could prevent thrombosis in HIT patients. They are presented in Table 1.

## 4. Conclusions

HIT II pathogenesis involves multi-step immune-mediated pathways dependent on the ratio of PF4/heparin and platelet, monocyte, neutrophil, and endothelium activation. For years, only platelets were known to take part in HIT II developing. A few years ago, specific receptors and signal-induced pathways in monocytes, neutrophils and endothelium were revealed. It had been shown that these cells that have become active released different newly founded compounds (platelet: TF, TNFα, NAP2, CXCL-7, ENA-78, platelet-derived microparticles; monocytes-TF-MPs; neutrophils-NETs), leading to additional cell activation and consequently thrombin generation resulting in thrombosis. Development of a new non-anticoagulant treatment for HIT II needs knowledge of FcγIIa receptors on platelets, monocytes, neutrophils and FcγIIIa on endothelium, chemokine (CXCR-2), and PSGL-1 receptors on neutrophils. IgG degradation, Syk kinase and NETosis inhibition are investigated in this field too. IdeS and DNases-related pathways should also be investigated for a better understanding of HIT pathogenesis and as possible HIT II treatment targets.

## Figures and Tables

**Figure 1 ijms-24-08217-f001:**
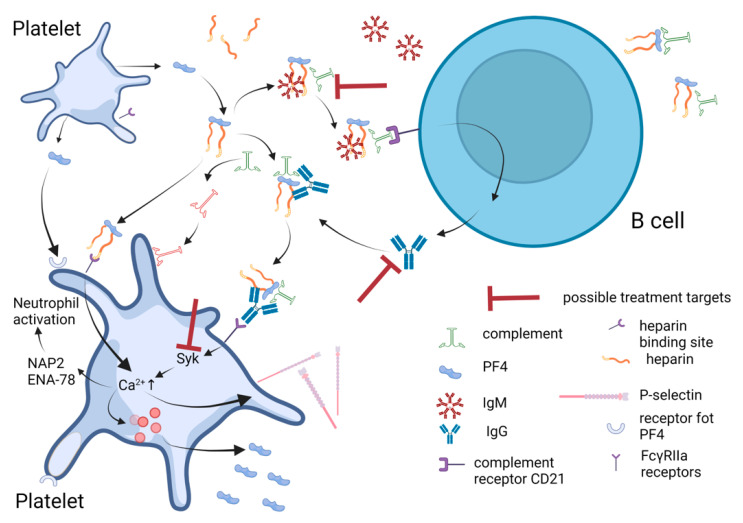
Platelets in HIT pathogenesis and possible treatment targets. PF4 levels in the blood increase after platelet activation. Cationic PF4 binds to anionic heparin. PF4-heparin complexes bind to platelets via the heparin-binding sites. PF4-heparin complexes activate complement and form complement deposition on the complexes, which leads to the binding to B cells by complement receptor 21 and results in antibody IgG synthesis. IgM takes part in complement activation too. PF4/heparin/IgM complexes play a role in the initiation of the immune response, activation of B cells and triggering of the production of specific antibodies (IgG). IgG bind to PF4-heparin-complement and form immune complexes. Immune complexes bind to platelets via FcγRIIa receptors leading to platelet activation. Platelet activation mediated by FcγRIIa receptors goes through Syk activation and Ca^2+^ concentration, increasing in the cytosol. Ca^2+^ ions induce the release of PF4 and other compounds from α granule leading to thrombosis (it is not shown here). Possible treatment targets are shown. ENA-78—epithelial-derived neutrophil-activating peptide 78, NAP-2—neutrophil-activating peptide 2, PF4—platelet factor 4, Syk—spleen tyrosine kinase. Created in Bio Render.com.

**Figure 2 ijms-24-08217-f002:**
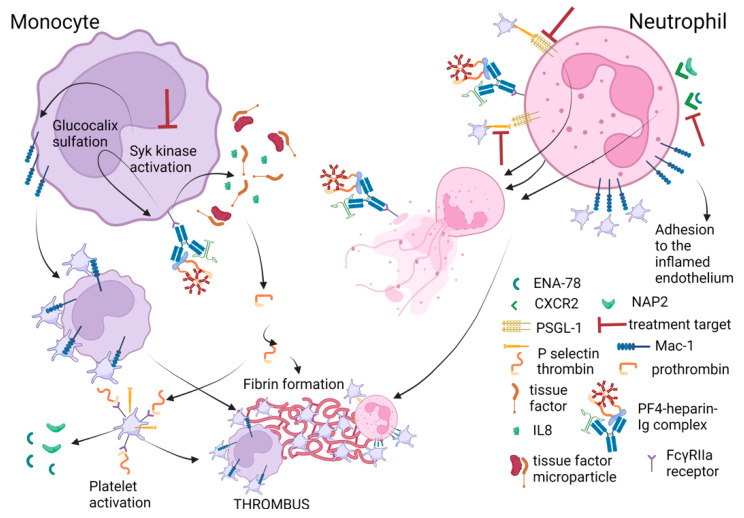
Cell (platelet, monocyte, and neutrophil) interaction in HIT and possible treatment targets. Neutrophils are activated through FcƴRIIa receptors. Activated neutrophils express Mac-1 and start NETosis. PSGL-1 and Mac-1 assist in the adhesion to platelets and neutrophil–platelet aggregate formation. Mac-1 additionally assists in the adhesion to inflamed endothelium. Activated platelets release NAP2 and ENA-78, leading to CXCR2-dependent neutrophil activation. Activated neutrophils form aggregates with platelets, release NETs and form complexes with PF4-heparin-Ig immune complexes. Monocytes release tissue factor microparticles, which activate prothrombin and initiate a coagulation cascade resulting in fibrin formation. Consequently, fibrin, platelets, and monocytes form a thrombus. Mac-1—oligodeoxynucleotide-binding protein, IL-8—interleukin 8, PSGL-1—P-selectin glycoprotein ligand-1, CXCR-2—chemokine receptor 2, ENA-78—epithelial-derived neutrophil-activating peptide 78, NAP2—neutrophil activating peptide 2, PF4—platelet factor 4. Created in BioRender.com.

**Table 1 ijms-24-08217-t001:** New possible treatment targets to manage HIT pathways.

Type of Cell	Possible Treatment Target	Research Achievements	Investigating Substance	Literature
Platelet	CD21(com-plement receptor)	Complement deposition on the complexes allows to bind to B cells by CR2/CD21.	-	[22,23]
	IgG	IdeS cleave the hinge region of heavy-chain IgG region (needed for interaction with FcγIIa) and disables platelet activation.	IdeS	[34]
	Syk kinase	Inhibition of Syk kinase disables platelet activation.	PRT318	[29,35]
Neutrophils	CXCR2	Neutrophil activation goes through CXCR2.	-	[40,48]
	NETosis	There are smaller venous thrombi, when neutrophils cannot undergo NETosis.	-	[28,40,50]
	DNases	PF4-NET antibody complexes are protected from degradation by NDases.	-	[40]
	CD62p	Induction of NETosis by HIT IgG pre-treated platelets depends on neutrophil–platelet association through P-selectin. Neutrophil–platelet aggregates in whole blood were significantly reduced by antibody blockage of P-selectin (anti-CD62p), PSGL-1 (anti-CD162) or FcγRIIA (IV.3 antibody), while the addition of DNase I or GSK484 did not affect cell–cell association	-	[28]
	PAD4	Absence of PAD4 or PAD4 inhibition with GSK484 abrogates thrombus formation but not thrombocytopenia. PAD4 inhibitor GSK48 and FcγRIIA neutralisation with IV.3 antibody strongly inhibited NETosis.	GSK48	[28]
Monocytes	Syk kinase	Inhibition of Syk kinase reduced thrombin and fibrin formation.	PRT318	[35,76]

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
