# Peer review of "Novel Knowledge about Molecular Mechanisms of Heparin-Induced Thrombocytopenia Type II and Treatment Targets"

_ijms, 2023, doi:10.3390/ijms24098217_

Round 1
Reviewer 1 Report
The review manuscript by MongirdienÄ— et al., entitled, Novel Knowledge about Molecular Mechanisms of Heparin-Induced Thrombocytopenia Type II and Treatment Targets contains many important information. The authors highlighted many interesting studies in the field. However, I think that the English of the manuscript needs to be revised. There are also several typo errors that need to be corrected.
1. Please modify this sentence and add reference.
Its frequency is much more common than type II: up to 10 % 38 (4) (I Ahmed, 2007)(2).
2. Please rephrase these sentences
This type will not be analysed there. HIT II developing involves an immune mechanism (7).
The review will summarize the nowadays knowledge about the pathogenesis of HIT II, new treatment being developed, and potential new therapeutic targets related to it.
It is known that HIT II develops because of platelet factor 4 (PF4)-heparin complex 60 formation leading to antibody synthesis (7).
3. Typo errors
Cationic PF4 binds to anionic heparin due to the electrostatic interactions (18). (19)
The classical pathway is one of them (22). IgM have been shown to take place in complement activation by classical pathway. )
It was later discovered that PF4 binds to glycosaminoglycans (GAGs) not only on platelets (36) but on monocytes (37), neutrophils (13) 38) and endothelial cells too (13).
4. No point at the end of the sentence
The immunogenicity of the PF4-heparin complexes depends on their net charge, and 126 a positive net charge was shown to facilitate the interaction with immune cells (21)
5. Grammatically not correct
The amount of PF4-heparin complexes formed was found to be maximal at equimolar PF4/heparin ratio (17,20).
The increased complement deposition on platelets could participate in platelet activation too (Figure 1)
The classical pathway is one of them (22). IgM have been shown to take place in complement activation by classical pathway. )
Complexes binding to platelets via FcÆ´RIIa resulted in thrombocytopenia which could occur in the absence of neutrophil extracellular traps – NETosis (28)
NET are extruded strands of DNA and granular material that serve critical functions in innate immunity by attracting pathogens (39).
6. Please rephrase, it is not well written
It was shown that disruption of IgM-C1 interaction could prevent PF4/heparin-mediated com-plement activation (23) and could be a therapeutic target in HIT II. It is not clear what concentration of the IgM and/or PF4-heparin initiates complement activation and why complement activation by PF4-heparin complexes varies widely among healthy persons.
5. If CD21 is expressed in platelets, please provide reference, or discuss this accordingly.
CD21(complement receptor) inhibition on platelets could be investigated as the possible treatment target too.
7. Please review the references in this part, they are provided in several brackets. Please revise the grammar
It is thought that neutrophils take part in the HIT’s induced thrombosis. Neutrophils were shown to promote thrombosis in HIT through formation of NET (28,43) (43,44)(44) and through FcγRIIa receptors.
Neutrophils have FcγRIIa receptors (45), (28,43) and activation through them (and through P-selectin) leads to generation of reactive oxygen species, degranulation (28,43), (32), (46) increased phagocytosis (44), integrin expression (38) 159 and formation of neutrophil-platelet aggregates (29).
It was shown that antibody connection to FcγRIIa receptors on neutrophils enhanced neutrophil aggregate formation through increased expression of macrophage adhesion molecule-1 (oligodeoxynucleotide-binding protein CD11b (Mac-1)) on the neutrophil surface (38) (Figure 2). Additionally, myeloperoxidase (MPO) levels were found to be higher in patients with HIT (29), 165 suggesting HIT association with neutrophil activation. Kandace Gollom with the research 166 group revealed steps in HIT by which neutrophils participate in thrombosis: 1) adhesion of stimulated neutrophils to inflamed endothelium, 2) the promotion of neutrophil migration into venular thrombi, 3) formation of antigenic PF4- (NET)-HIT antibody complexes 169 (40) (43,44).
Neutrophil can adhere to endothelium because of some reasons. The first reason could be that HIT antibody exposure increases the Mac-1 expression on the surface of neutrophils (38). The second one is believed to be the growing venular injury because the increased turbulent flow around thrombi enhances neutrophil adhesion to the endothe ium (38). The third, endothelial cells activation by HIT antibodies increases the expression of P and E selectins (47). During thrombus formation, PF4 release from platelets within the thrombus could lead to increased assembly of antigenic complexes, resulting in more HIT antibody binding to endothelial cell and its activation (13). A
Smaller venous thrombi formation was shown in HIT mice models, when neutrophils could not undergo NETosis but enhanced arterial clots, and thus thrombocytopenia developed (40). So, DNase treatment could prevent the venular thrombus size increase and prothrombotic state in HIT. It was shown that neutrophil migration into venous thrombi could be stopped with the blockade of CXCR2 (40) . But NAP2 and ENA- 78 contribution to the chemo gradient should be investigated deeper.
It was found that immune complexes (ICs) could activate neutrophils in HIT too. Activation goes via FcÆ´RIIa: 1) directly, and 2) by platelet activation leading to P-selectin and P-selectin glycoprotein ligand-1 (PSGL-1) release from activated platelets resulting in 217 neutrophil activation (28,43) . Despite P-selectin/PSGL-1 interaction not being the only mediator of neutrophil/platelet interaction (50), (40) blockade of P-selectin (CD62p) or PSLG-1 (CD162) markedly inhibit neutrophil/platelet interaction induced by HIT antibod- 220 ies and prevent DNA release by neutrophils in the presence of HIT ICs activated platelets (28).
Summarizing the importance of neutrophils in the course of the HIT, it can be said that neutrophils promote thrombosis through NET formation and endothelium activation. Activated neutrophils release NETs, that consequently activates prothrombin and leads to thrombosis
8. Please review the references in this part, they are provided in several brackets. Please revise the grammar
Monocytes are shown to be activated by P-selectin expressed on platelet . It should be mentioned that TF generated from monocyte activates platelet protease-activated receptor (PAR-1) (1). Platelet activation through PAR-1 and FcÆ´RIIA (56) (or thrombin and collagen, or thrombin and FcÆ´RIIA ) leads to generation of subpopulation of highly activated platelets named coated platelets (1). Monocytes activation by HIT ULICs leads to Mac-1 expression leading to platelet-monocyte aggregation (32). It should be mentioned that IdeS treatment abolished the increase of TF expression induced by HIT anti bodies (33) and could also reduce thrombotic risk . Summarising, monocytes can be activated by PF4-heparin antibody complexes through FcyR IIa receptors (leading to Mac-1 expression resulting in platelet-monocyte aggregation), platelet expressed P-selectin, or through connection to GAGs on their surface. Activated monocytes produce TF-MPs and other compounds related to the develop- ment of thrombosis. Monocyte activating pathway through FcyR IIa receptors was shown to be realized through Syk kinasesSo, Syk kinase inhibition could be useful for avoiding thrombosis by acting on both monocytes and platelets..
It is believed that IVIG inhibit platelet and other cells activation through FcγIIa receptors. Several studies have shown that the mechanism of action of IVIG is largely due to its competition with anti-PF4/H to bind to the FcgRIIA receptor and this may differ depending on the FcgRIIA H31R polymorphism, but exact mechanism of IVIG remains unknown (75).. Cangrelor was shown to be useful in cardiovascular surgery when PF4-heparin antibodies are present and intraoperative heparin use is mandatory (66, 73), but treatment efficacy must be assessed for each patient with a functional assay before its use (73). IdeS has not yet been studied in HIT patients.
Knowledge about FcγIIa receptors on platelets, monocytes, neutrophils and FcγIIIa on endothelium, chemokine (CXCR-2), and PSGL-1 receptors on neutrophils could allow to develop a new non-anticoagulant treatment for HIT II. IgG degradation, Syk kinase and NETosis inhibition are in the field of developing new treatment possibilities too. Accordingly, IdeS, and DNases related pathways should be investigated for better understanding of HIT pathogenesis and possibilities to be the HIT II treatment targets.
Author Response
Thanks to the Reviewer for carefully reviewing the article and helpful comments. We have corrected the article in the light of the comments. It improved the quality of the article. The following corrections have been made please see in the atachment.

Reviewer 2 Report
The authors prepared a review article on HIT II, for which new findings have been reported in recent years, based on many references. As a reviewer, I found this review to be a very good review, with a structure that includes summaries of complex content for each unit in order to make it easier for readers to understand.
I would like to ask the author if he/she intends to 1) underline the words on lines 190 and 191 and 2) change the typeface of the text on lines 357-359.
Author Response
Thanks to the Reviewer for carefully reviewing the article and helpful comments. We have corrected the article in the light of the comments. The following corrections have been made.
I would like to ask the author if he/she intends to 1) underline the words on lines 190 and 191 and 2) change the typeface of the text on lines 357-359.
The words on lines 190 and 191 have been underlined.
The typeface on lines 357-359 have been changed.

Round 2
Reviewer 1 Report
The authors addressed the comments of the reviewer.